# Peer review of "Dispersal and Repulsion of Entomopathogenic Nematodes to Prenol"

_biology, 2019, doi:10.3390/biology8030058_

Round 1

Reviewer 1 Report

A follow up, comparative study that complements work on phenol published by the authors in 2017.

Specific comments to be addressed:

Abstract 

Line 16: 'volatile odor' is tautology; it would also be useful to define the type of compound here, e.g. 'Prenol is a volatile, natural alcohol...'

Line 17: it would be more accurate to say ..'this odorant...' rather than 'this odor...'

Introduction

Line 29: it is incorrect to refer to nematode juveniles as larvae; nematodes have juvenile stages, which resemble the adult structurally, and unlike larvae (e.g. in some insect orders) do not metamorphose into adult stages; suggest...'modified third stage juvenile, the infective juvenile (IJ) stage....'

Line 43: as in line 17...'Prenol is an odorant...' rather than 'odor'

Materials and Methods

Line 100: presumably prenol was obtained as >99% pure [as in Ref 12]? The purity should be included here.

Line 101: it is not clear why '200 mmol dilutions' of prenol are mentioned when chemotaxis experiments (Fig 1) used dilutions starting from 2 M prenol?   

Line 101: the solubility of prenol in water at 20C is about 2 M (170g/l); the initial dilution from pure prenol (above) to 2 M must have been in a solvent miscible with water, presumably ethanol [?] [as in Ref 12]; if so, this should be made clear in section 2.2. 

It may be implied from Line 108, that subsequent serial dilutions were in Milli W water [?]- if so, this should also be made clear in section 2.2.

Assuming serial dilutions from 200 mM prenol were all made with Milli Q water [?], there would have been a diminishing amount of ethanol in the different prenol solutions tested from ca 80% ethanol for 2 M prenol; ca 8% ethanol for 200 mM prenol; down to ca 0.0008% at 20 um prenol. 

If this is the case, were control experiments conducted to check that this did not influence the results of the dose response chemotaxis experiments shown in Fig 1?

Line 105: refers to reference 12 for details of the chemotaxis assay; ref 12 in turn refers to other papers; a primary reference source for the assay method should be substituted.

Sections 2.3, 2.4: 

There is no mention of the temperature [room temperature?] under which assays were conducted; this information should be added [also for sections 2.5, 2.6]

In previous chemotaxis work [Ref 18] a threshold of 3 or more IJ in a scoring circle was used - was this the case in the present work [?]; this information should be added.

What controls were used? Did IJs show any response? [see Line 101 above]

Line 119: refers to checking the (level of) 'participation of IJs' on chemotaxis plates; this information is not mentioned in the Results section?

Section 2.6:

What controls were used?

Results

Line 208: add conc. of prenol [200 mmol?]

Fig 1: x-axis 2 uM should be 2 mM

Fig. 1: the response of S. riobrave and S. glaseri to prenol repeats experiments in Ref 12; the dose response of riobrave to prenol is similar to the earlier study but the response of glaseri appears to differ to some extent - is there a possible explanation(s) for this - see Discussion?

Fig. 2: please confirm the mean and variances (SEM) shown for S. carpocapsae (D) at 4h and 7 days are not significantly different?

Lines 224 (Fig 2 legend): add dose of prenol used [200 mM?]

Line 252: add conc. of prenol [200 mmol?]

Line 266: add conc. of prenol [200 mmol?]

Discussion:

Lines 278-281: see comments re Fig. 1 above

Author Response

We appreciate the careful reading of the manuscript and the suggestions provided by the reviewer. We have changed the language in all the ways that they have suggested and have added details to clarify information, fix Fig. 1 and address the majority of their concerns.

Regarding the concern about the way in which prenol was prepared, the dilutions of prenol (even at 2M) were done in milliQ water. We have added some language to make this clarification. Since this was the case, we do not feel there was any need for a control experiment to evaluate the influence of ethanol (since no ethanol was used). We have also added in a description of homogenization of 2M prenol (to ensure that the mixture is homogenous— incase some of the prenol did not dissolve fully into the water at this high concentration.

Regarding participation we have added the detail in the methods that this information is presented in the supplemental figure 3. We are of the opinion that the participation data is present merely to show that the CI is representative of a substantial proportion of the population, due to the complexity of the information we felt that it was better left in the supplementary and not made a main focus in the paper.

We have addressed the concerns by reviewer 1 regarding comparison of S. glaseri responses to the dose response curve compared to the previous study, additional language was added to the discussion to compare the results and make a conjecture about the reasons why the behavior might vary (mainly due to natural variations in behaviors, as well as the differences in age ranges that were allowed for the experiments in this study compared to the study by Baiocchi et. al. 2017). Regarding the concern of Figure 2 we have double checked our PRISM program and re-ran the ordinary one-way ANOVA for the chemotaxis index results of S. carpocapsae the statistical analysis still shows that there is no significant difference between 4 hours and 7 days post-collection. If we run a t-test we find statistical significance, but since the one-way ANOVA with the Tukey multiple comparisons test is the more rigorous and reliable test, this is the test we chose to use. We also made note that the ANOVA indicated that there was no significant difference in standard variations across the categories tested (4-hour, 1 day and 7-day post-collection time points).  Please note that the error bars shown are SEM and for standard deviation they are somewhat larger and the standard deviation error bars for the 4 hour and 7-day time point do overlap.

Reviewer 2 Report

This paper demonstrates the effect of phenol on the dispersal and repulsion of entomopathogenic nematodes. It is well written and the materials and methods and analysis of results are scientifically sound. The results are presented clearly and the discussion is appropriate.

A few suggestions may be considered by the authors:

Page 1, line 22: ital "Steinernema"

Line 27: delete "which"

Line 29: hyphenate "third-stage"; delete "larva" or replace with "juvenile" to be consistent in word usage

Line 32: make plural "stresses"

Line 33 and throughout: I prefer to use the Columbia comma to increase clarity

Page 2, line 43: add taxonomic authority and date of publication after the first use of a latinized binomial of an insect name according to the rules of zoological nomenclature

Line 44-46: see above

Line 58: lowercase "magentotaxis"

Line 78: capitalize "White"

Line 85-86 and 89-91: ml

Page 3, line 100: Commonly known as prenol, 3-methyl-2buten-1-ol, was obtained....

line 141: replace "placed under" with "photographed with"

Page 7, line 248: capitalize "Measure"; ital "two-way ANOVA" and "Tukey's multiple comparison test" ; spellcheck "teste"

Line 249: "species'  "

Author Response

      We appreciate the careful reading of the manuscript and the suggestions provided by the reviewer. We have made all the text changes that were suggested and have added in the taxonomic authority information for the species mentioned. Regarding the comment on Line 33 and throughout the manuscript about the use of the Oxford comma, we think this is what the reviewer meant as we couldn’t find any information on anything called a “Columbia comma.” This is a writing style choice and not something that affects the clarity or science described in the manuscript. We would prefer to keep the manuscript as is, without the Oxford comma.